# Screening and Verification of Photosynthesis and Chloroplast-Related Genes in Mulberry by Comparative RNA-Seq and Virus-Induced Gene Silencing

**DOI:** 10.3390/ijms23158620

**Published:** 2022-08-03

**Authors:** Yong Li, Cui Yu, Rongli Mo, Zhixian Zhu, Zhaoxia Dong, Xingming Hu, Wen Deng, Chuxiong Zhuang

**Affiliations:** 1College of Life Sciences, South China Agricultural University, Guangzhou 510642, China; liyong8057@hbaas.com; 2Cash Crops Research Institute, Hubei Academy of Agricultural Sciences, Wuhan 430064, China; yucui_b2020@163.com (C.Y.); monamorus@hbaas.com (R.M.); zhuzhixian@hbaas.com (Z.Z.); dongzhaoxia9@hbaas.com (Z.D.); 13607121598@163.com (X.H.)

**Keywords:** photosynthesis regulation, mulberry, transcriptomes, re-annotation, VIGS

## Abstract

Photosynthesis is one of the most important factors in mulberry growth and production. To study the photosynthetic regulatory network of mulberry we sequenced the transcriptomes of two high-yielding (E1 and E2) and one low-yielding (H32) mulberry genotypes at two-time points (10:00 and 12:00). Re-annotation of the mulberry genome based on the transcriptome sequencing data identified 22,664 high-quality protein-coding genes with a BUSCO-assessed completeness of 93.4%. A total of 6587 differentially expressed genes (DEGs) were obtained in the transcriptome analysis. Functional annotation and enrichment revealed 142 out of 6587 genes involved in the photosynthetic pathway and chloroplast development. Moreover, 3 out of 142 genes were further examined using the VIGS technique; the leaves of *MaCLA1*- and *MaTHIC*-silenced plants were markedly yellowed or even white, and the leaves of *MaPKP2*-silenced plants showed a wrinkled appearance. The expression levels of the ensiled plants were reduced, and the levels of chlorophyll b and total chlorophyll were lower than those of the control plants. Co-expression analysis showed that *MaCLA1* was co-expressed with *CHUP1* and *YSL3*; *MaTHIC* was co-expressed with *MaHSP70*, *MaFLN1*, and *MaEMB2794*; *MaPKP2* was mainly co-expressed with *GH9B7*, *GH3.1*, and *EDA9*. Protein interaction network prediction revealed that MaCLA1 was associated with RPE, TRA2, GPS1, and DXR proteins; MaTHIC was associated with TH1, PUR5, BIO2, and THI1; MaPKP2 was associated with ENOC, LOS2, and PGI1. This study offers a useful resource for further investigation of the molecular mechanisms involved in mulberry photosynthesis and preliminary insight into the regulatory network of photosynthesis.

## 1. Introduction

Mulberry (*Morus alba* L.) is a fast-growing perennial shrub that belongs to the Moraceae family and is widespread in China. Mulberry has always been known as a traditional silkworm feed and has important economic value in the field of silkworm farming. Mulberry leaves are rich in many nutrients, such as vitamins and minerals. In addition, mulberry leaves also possess many medicinal effects due to their numerous bioactive compounds, such as steroids, terpenes, alkaloids, and flavonoids. Therefore, on the one hand, mulberry is widely used in food, medicine, feed, and ecology, and mulberry is also gradually attracting wide attention due to its extremely high nutritional and medicinal value. Mulberry leaves and mulberry have been listed by the Ministry of Health as “one of the agricultural products that are both food and medicine”. On the other hand, mulberry is highly exposed to stress tolerances, such as drought [1], salinity [2], heavy metal stress [3,4], waterlogging [5,6], and resistance to *B. cinerea* [7]. As a result, many studies have been conducted to investigate the mechanism of mulberry response to abiotic and biotic stress using the transcriptome and proteome [2,8].

Virus-induced gene silencing technology (VIGS) is an RNA-mediated post-transcriptional gene defense mechanism. Compared to RNA interference and Agrobacterium-infected genetic transformation methods, VIGS can quickly silence endogenous genes without a stable transformation system, and a short sequence is enough to silence the target gene with the simple operation process. Therefore, VIGS technology has been widely used in functional genomics studies of many species, which mainly degrades gene expression by targeting gene fragments into viral vectors to mediate RNA interference in plants [9]. Besides a detailed study of gene function in model plants like tobacco, *Arabidopsis*, and tomato [10], the tobacco rattle virus (TRV)-VIGS system was also very powerful for use in some species whose stable genetic transformation systems are not yet established, such as pepper and mulberry, to explore the function of the interesting target gene. Tomatoes, as a typical model plant, had their fruits yellowed and their lycopene levels significantly reduced after TRV-VIGS knocked down the *SlNAP7* gene [11]. However, a previous study showed decreased levels of capsaicin and dihydrocapsaicin in pepper by TRV-VIGS to reduce *AT3* gene expression [12]. Because of the genetic transformation system, the major limitations for gene function studies in mulberry are still being investigated. Fortunately, VIGS is a very powerful technology. TRV-VIGS was used to silence the protein kinase gene (*MmSK*) in mulberry. The plants with decreased *MmSK* gene expression were under drought stress. The soluble protein and proline content, the activities of superoxide dismutase (SOD), and peroxidase (POD) in the plant were reduced to varying degrees [13]. Furthermore, TbCSV-VIGS was used to induce *MaMET1* gene expression silencing while enhancing the expression of resistance genes to increase resistance to B. *cinerea* in mulberry [7]. Nevertheless, the application of VIGS in mulberry studies is still rare.

Photosynthesis is very important for a plant’s growth and productivity. As a typical C3 plant, the photosynthetic capacity and chlorophyll metabolism of the leaves are normally influenced by a variety of factors such as abiotic and biotic stress, plant developmental stages, and plant genotypes. Currently, most research has focused on the effects of stress and artificial cultivation techniques on the photosynthetic properties of the mulberry. For example, chlorophyll fluorescence parameters remain at a steady-state level under continuously high flooding [6]; of three mulberry genotypes under water stress, Selection-13 showed significantly higher net photosynthesis rates (Pn) and hydraulic conductivity of stem and leaf than Kollegal Local, and Kanva-2 [14]. With continued exposure to elevated atmospheric CO_2_ concentration for the drought-tolerant Selection-13 and susceptible Kanva-2, Selection-13 showed superior photosynthetic performance [15]. Increased atmospheric CO_2_ concentration enhanced carbohydrate and N compound accumulation in mulberry leaves, which affected biomass production and nutrients [16]. Moreover, mulberry (*Morus alba* L.) showed a high tolerance to Cd with normal photosynthesis and antioxidant protection [4]. However, few studies focus on comparing the photosynthetic ability of mulberry genotypes with different yields under natural growing conditions to unveil the expression regulation of key genes. 

On this basis, we investigate the mechanism of photosynthesis more systematically and deeply for the yield differences of mulberry genotypes under natural growth conditions. Here, we mainly selected the two high-yielding (E1; E2) and one low-yielding (H32) mulberry genotypes for the determination of two-time points (10:00 and 12:00). In our study, we identified differential expression genes (DEGs) by comparative transcriptome in E1, E2, and H32 at 10:00 and 12:00. DEGs were then subjected to GO and KEGG analysis. Furthermore, we screened 142 photosynthesis- and chloroplast-related DEGs, from which we finally selected three important DEGs for functional verification by VIGS and then to further analyze the regulatory network of gene co-expression of mulberry photosynthesis. This study can provide favorable data support for the functional analysis of other important photosynthetic genes in the later stage and show the direction for the study of the photosynthetic regulation network, laying the foundation for mulberry molecular breeding.

## 2. Results

### 2.1. Transcriptome Sequencing and Gene Structure Re-Annotation

Mulberry is a typical C3 plant. We first measured the net photosynthetic rate (Pn) for two high-yielding (E1, E2) and one low-yielding (H32) mulberry genotypes and found it to be a typical bimodal curve with an obvious photosynthesis lunch break. The first peak that appeared at 10:00 was significantly higher than the second peak that appeared at 14:00, and the valley at 12:00 is the photosynthetic lunch break point (Appendix A). Therefore, we performed three mulberry genotypes E1, E2, and H32 with three biological replicates at 10:00 and 12:00 time points for transcriptome sequencing, which are helpful to understand the expression regulation of key genes related to photosynthesis and to study the mechanism of mulberry photosynthesis. In total 18 cDNA libraries were constructed, and about 8.4 Gb raw data were generated in every library. After quality control, a total of 148.3 Gb of clean data were obtained. The clean reads accounted for 98.1% of the total raw reads, and the percentage of Q20 and Q30 bases reached about 98% and 95%, respectively. The GC content of all samples was 45–47% (Appendix A). The above results showed that the sequencing quality was good, which can be used for subsequent analysis.

We found that when using wild mulberry (*Morus notabilis*) as the reference genome [17], the alignment rate was only 70% (Appendix A), while when using cultivated mulberry (*Morus alba*) as the reference genome [18], the alignment rate was over 90% (Table 1). However, we could not retrieve the gene structure annotation information for cultivated mulberry, so we re-annotated the cultivated mulberry genome using 18 transcriptome data in this study based on the *Arabidopsis* Araport11 annotation pipeline [19]. The newly annotated cultivated mulberry genome contained 22,664 protein-coding genes, of which 20,185 genes had functional annotations, and the integrity reached 93.4% in that evaluation by BUSCO (Table 2), which has reached the high-quality gene annotation standard. The newly annotated genes provide a high-quality research basis for our subsequent transcriptome quantification, differential analysis, and gene cloning.

### 2.2. The Identification of Differentially Expressed Genes 

Gene expression was quantified in each transcriptome dataset using our newly annotated gene structure. To ensure the accuracy of subsequent analysis results, we used Principal Component Analysis (PCA) to identify plausibility among biological replicates. We found that the PC1 value could represent 80.9% of the sample information (Appendix A), showing that the repeatability of the sample in this study was great for subsequent analysis. Then we performed difference analysis and a total of 6587 differentially expressed genes (DEGs) were obtained. We visualized DEGs by heatmap and found that genes with high expression in E1 and E2 but very low expression in H32 were clustered together (Appendix A). In addition, we compared the DEGs of different cultivars or time points, respectively, and obtained 7 groups of DEGs. The number of DEGs was significantly higher than that of the same cultivars at different time points, and the E2-10_vs_H32-10 group had the largest number of DEGs (Figure 1 and Appendix A). 

The overlapping DEGs between the seven differential analysis groups were obtained by the Venn diagram. Comparing high-yielding cultivars (E1 and E2) with low-yielding cultivars (H32, set as control) at the same time point, we found a total of 1345 DEGs in the four groups (Figure 2A), and 151 DEGs were found in the three groups of same cultivars at different time points (Figure 2B). The above results showed that the number of DEGs in different cultivars is greater than the number of DEGs in different periods of the same cultivar.

### 2.3. GO and KEGG Annotation of DEGs 

To further investigate the DEG functions, we performed Gene Ontology (GO) function annotation and enrichment analysis. GO terms with a *p*-value ≤ 0.05 were selected as significantly enriched and 624, 126, and 310 GO terms in biological processes, cellular components, and molecular function, respectively (Appendix A). Through the above analysis, we selected the top 10 GO terms for each component among all the significantly enriched GO terms for visualization. As shown in Figure 3 and Appendix A, DEGs were involved in biological processes including defense responses, responses to various strong light conditions, no light, heat and other stresses, responses to salicylic acid stimulation of cells, RNA editing and modification, chloroplast RNA processing, and modification and chloroplast-signaling pathway. In cellular components, DEGs were significantly enriched in the chloroplast, chloroplast envelope, stromal thylakoid and chloroplast thylakoid membranes, as well as mitochondrial and plastid plasma membranes. In terms of molecular function, DEGs have been enriched mainly with ATP binding, RNA binding, oxygen, iron, magnesium, ion binding, various enzyme and transferase activities, carbohydrate binding, and various photosynthesis-related enzyme activities. The above results suggest that these DEGs may be involved in the photosynthetic process through carbohydrate metabolism processes, photosynthetic electron transport in photosystem II, negative regulation of catalytic activity, iron ion binding, and hydrolase activity pathways.

To thoroughly investigate the biological functions, we performed KEGG analysis on all DGEs. These DEGs were involved in a total of 137 pathways, of which 32 pathways were significantly enriched and 6–8 pathways were significantly enriched in each group (Figure 4 and Appendix A). The main significant pathways are photosynthesis, carbon fixation, plant hormone signal transduction, circadian rhythm, plant-pathogen interaction, and secondary metabolic pathways. 

### 2.4. The Change of Chloroplast- and Photosynthesis-Related Genes Expression 

A total of 142 DEGs related to photosynthesis and chloroplast development were identified in the study (Appendix A). Among them, 106 DEGs were obtained simultaneously from high-yielding and low-yielding mulberry at the same time points, while the expression levels of 13 DEGs changed significantly in all four groups (Appendix A). When comparing the two-time points in the same cultivars, the number of DEGs was relatively small; 74 DEGs were scanned, and a total of 13 DEGs had changed significantly in all three groups (Appendix A).

We would like to further elucidate the function of these genes, *Moa02g03470.1* shared 87% homology with *AT4G15560.1*, which encodes the first enzyme of the 2-*C*-methyl-D-erythritol-4-phosphate pathway. CLA1 (1-deoxy-D-xylulose-5-phosphate synthase) was mainly expressed both in photosynthetic and non-photosynthetic developmental tissues in *Arabidopsis.* Mutational *CLA1* disrupted the normal development of chloroplasts, resulting in plant albino [20,21]. *Moa02g03470.1* was designated as *MaCLA1*. *MaCLA1* expression was significantly up-regulated in E2-12_vs_H32-12 and E1-12_vs_E1-10 (Appendix A). *Moa06g10020.1* has 86% homology with *AT2G29630*, which resides in the chloroplast matrix as Thiamine pyrimidine synthase (THIC) (Table 3). Chloroplast development was delayed in an *Arabidopsis* mutant *thiC*, resulting in pale leaves and lethal seeding [22]. In our study, the expression of *MaTHIC* (*Moa06g10020.1*) was significantly up-regulated in different periods of the same cultivars (Table 3). *Moa11g02720.1* shared a similarity of 84.9% with *AT5G52920.1*, which encodes a pyruvate kinase subunit. Chloroplast pyruvate kinase (PKP2) affected seed germination, which in turn affected plant growth and development [23]. *MaPKP2* (*Moa11g02720.1*) was expressed significantly up-regulated in E2-10_vs_H32 (Appendix A). 

The expression level of some genes belonging to the PPR family also changed significantly. The expression of *Moa07g15800.1* was significantly increased at 12:00 compared to 10:00 in the same cultivars (Appendix A), and its *Arabidopsis* homolog, AT3G18110, named *AtEMB1270* or *AtACM1*, which is crucial for chloroplast development, resulting in cotyledonal albino in *Arabidopsis* after knockout [24]. The expression levels of *MaOTP80*, *MaOTP81*, *MaOTP82*, *MaOTP84*, and *MaOTP85* were all up-regulated, among which *MaOTP85* was significantly up-regulated in different cultivars and two-time points (Appendix A). In previous studies, *MaOTP85* are chloroplast editing factors that directly or indirectly impact chloroplasts [25]. Furthermore, the expression levels of *FLN1*, *FLN2*, and *TRXz*, which regulate the transcriptional activity of plastid-encoded RNA polymerase (PEP), were significantly altered in this study (Appendix A). The *FLN1*, *FLN2*, and *TRXz* mutants caused plants to be yellow or albino, which seriously affects plant growth and development in *Arabidopsis*, rice, tomato, and pepper, showing that their functions are highly conserved in plants [26,27,28].

Numerous genes involved in the photosynthetic system have significantly changed in their expression. As homologous genes of *Arabidopsis* ATCG00490.1, *pgp056* was significantly down-regulated in E2-10_vs_E2-12, which annotated as ribulose-bisphosphate carboxylases (*RbcL*), (Table 3 and Appendix A). Pgp064 was significantly up-regulated in E2-12_vs_H32-12 and E2-10_vs_E2-12, sharing 97% homology with *Arabidopsis psaA* gene, encoded PSI protein (Table 3 and Appendix A). *Pgp068* was significantly up-regulated in E2-12_vs_H32-12 and E2-10_vs_E2-12, shared 97% homology with *Arabidopsis psbC* gene, encoded the PSII CP43 reaction center protein (Table 3 and Appendix A). 

### 2.5. Silencing of Candidate Genes Altered Leaf Phenotypes of Mulberry

To verify whether the candidate genes are involved in the photosynthetic process and then affect the growth and yield of the mulberry, we selected three genes (*MaCLA1*, *MaTHIC*, and *MaPKP2*) from 142 DEGs related to photosynthesis and chloroplast development to verify the function by VIGS technology. The VIGS system includes two virus strains, TRV1, and TRV2, for which the TRV2 vector map information has been shown in Appendix A. Appendix A was the DNA marker indicating stripe size. The pTVR2-*MaPDS* vector was used as a positive control, which theoretically results in an albino phenotype, and the pTRV2 empty vector (EV) was used as a negative control. The silencing plants began to appear in phenotypes after 15 days. The pTRV2 (EV) plants returned to normal growth at a later time point, essentially corresponding to wild-type (WT) (Figure 5A). The leaves and stems of pTRV2-*MaPDS* plants appeared albino, significantly affecting plant growth (Figure 5B,C). The leaves of pTRV2-*MaCLA1* and pTRV2-*MaTHIC* silenced plants showed yellow or even white pigmentation (Figure 5D,E). The leaves of pTRV2-*MaPKP2* silenced plants showed apparent shrinkage (Figure 5F). We observed that the plant phenotype gradually spread along the leaf veins, which was mainly related to the spread of the virus mainly via the leaf veins (Figure 5G).

To further confirm the above result, we examined expression levels and chlorophyll content for silenced plants. Compared to the control pTRV2 (EV) plants, expression levels of *MaCLA1*, *MaTHIC*, and *MaPKP2* were all down-regulated in silenced plants (Figure 5H), showing that the expression of three genes was silenced to varying degrees. Silenced, these genes cause the mulberry leaves to turn yellow, allowing pigment content to be altered. Therefore, we simultaneously measured the chlorophyll content in the leaves of VIGS-inoculated mulberry plants. The chlorophyll a did not change, but the chlorophyll b and total chlorophyll content were lower than the control plants (Figure 5I). The results indicated that the pigment content in the leaves was induced by decreased gene expression, which in turn caused the leaves of mulberry plants to turn yellow.

### 2.6. Gene Co-Expression and PPI Networks of Candidate Genes

To better understand the network pattern of gene expression regulation, we constructed a gene co-expression and PPI (the Protein-Protein Interaction) network. The results showed that *MaCLA1*, *MaTHIC*, and *MaPKP2* were co-expressed with 3, 59, and 23 genes, respectively (Figure 6). *MaCLA1* was co-expressed with *MaYSL3* (iron transporter), *MaBOU* (amino acid transporter), and *MaCHUP1* (encoding a chloroplast-localized protein) (Appendix A). *MaTHIC* was co-expressed with *FLN1* and most PPR genes, including *EMB2794* and *EMB3120* (Appendix A); *MaPKP2* was mainly co-expressed with *GH9B7* (glycosyl hydrolase 9B7), *GH3.1* (encoding an IAA-amide synthase-like protein), and *EDA9* (a 3-phosphoglycerate dehydrogenase) (Appendix A). These genes are all involved in photosynthesis and chloroplast development in *Arabidopsis* and can affect plant growth and development [29,30,31,32,33]. Furthermore, the PPI network was used to speculate on the possible interaction proteins for candidate genes. CLA1 is associated with RPE (Ribulose-phosphate 3-epimerase; Chloroplastic), GPS1 (Solanesyl diphosphate synthase 3), and DXR (1-deoxy-D-xylulose 5-phosphate reductoisomerase). THIC interacts with THI1 (Thiamine thiazole synthase 2), PUR5 (Phosphoribosylformylglycinamidine cyclo-ligase), and TH1(Thiamine biosynthesis bifunctional protein ThiED). PKP2 interacts with ENOC (Cytosolic enolase 3), PDH-E1_ALPHA (Pyruvate dehydrogenase e1 component subunit alpha), and PGI1 (Glucose-6-phosphate isomerase). All of these genes are related to chloroplast and photosynthesis according to the annotated function and share the most commonly interacting proteins with *Arabidopsis* (Appendix A; Appendix A). These results indicate that *MaCLA1*, *MaTHIC*, and *MaPKP2* regulate photosynthesis and chloroplast development in mulberry by working in concert with multiple functionally similar genes.

## 3. Discussion

### 3.1. The Highly Quality Genes Annotation of Cultivated Mulberry (Morus alba)

High-quality gene annotation for reference genomes is a powerful tool for molecular genetic studies of mulberry. *Morus notabilis* is a wild mulberry species whose genome was reported in 2013 [17], while *Morus alba* is a cultivated mulberry species whose reference genome has reached high-quality and chromosome-level reported in 2020 [18]. However, we could not retrieve the gene structure annotation information about cultivated mulberry (*Morus alba*), so we re-annotated the cultivated mulberry genome using our 18 transcriptomes data based on the *Arabidopsis* Araport11 annotation pipeline. In our study, we reannotated 22664 protein-coding genes, of which 20,185 genes had functional annotations, a mean CDS length of 1226 bp, a mean exon number of 5.07, and the annotated genes with 93.4% of BUSCO (Table 2). High-quality gene annotation for reference genomes will provide a high-quality research basis for our subsequent transcriptome quantification, differential analysis, and gene cloning.

### 3.2. Photosynthesis- and Chloroplast-Related DEGs Affect Photosynthesis to Further Change Biomass and Yield in Mulberry

Photosynthesis is a complex life process that includes dozens of chemical reactions that occur sequentially in living organisms. Many studies have been conducted to increase plant biomass and yield by improving photosynthetic capacity and efficiency [34,35]. However, the result of gene expression regulation is one of the factors affecting photosynthesis, which further affects plant growth and development. Therefore, we identified the key genes controlling mulberry photosynthetic capacity and efficiency based on transcriptome analysis of high- and low-yielding mulberry cultivars. Longer goals are the breeding of high-yielding mulberry varieties through stable genetic transformation. A total of 6587 DEGs were obtained in our study, of which a total of 142 photosynthesis- and chloroplast-related genes were differentially expressed, including 106 genes in different cultivars and 74 genes in different periods (Appendix A). The expression levels of genes encoding the core proteins of the photosynthetic system were altered, affecting the rate of photosynthesis [36]. Overexpression of *PsbS* in rice enhanced plant photosynthetic capacity to increase field biomass and grain yield [37]. 

Mutated *CLA1* in *Arabidopsis* showed that the leaf’s albino phenotype interfered with the development of chloroplasts, and the function of *CLA1* was highly conserved in plant evolution [20,21]. *Moa02g03470.1* is highly homologous to AT4G15560.1 (*CLA1*) (Table 3). Moreover, the expression of *MaCLA1* was significantly up-regulated in E2-12_vs_H32-12 and E1-12_vs_E1-10 (Appendix A). We speculated that the high expression of *MaCLA1* in E2 resulted in a high yield. The *MaTHIC* is located in the chloroplast. The leaves of the *thiC* mutant in *Arabidopsis* are pale, and the seedlings are lethal, but thiamine supplementation can restore and maintain their normal growth [22]. *MaTHIC* shares a high degree of homology with the *Arabidopsis* gene, and *MaTHIC* expression was significantly up-regulated in different periods of the same cultivars (Appendix A), so *MaTHIC* may also share the same function as *Arabidopsis*. *MaPKP2* was significantly up-regulated in E2-10_vs_H32 (Appendix A), and it was the important chloroplast pyruvate kinase that influenced *Arabidopsis* development [23].

Most PPR proteins have different effects on plants. Mutations in most PPRs proteins result in physiological defects that contain missing pigment that cause further leaf albino or yellowing, photosynthetic defects, and plant growth restriction [38,39]. PPR proteins are involved in post-transcriptional regulation of chloroplast or mitochondrial, such as RNA maturation, editing, splicing, etc., and the synergistic effect of RNA metabolism often affects the function of chloroplast and mitochondria, which affects photosynthesis, respiration, and development [40,41]. The expression of most genes encoding PPR proteins changed significantly in our study (Appendix A), such as *Moa07g15800.1 (MaEMB1270)* (Table 3, Appendix A), the knockout of which in *Arabidopsis* damages abnormal chloroplast development, causing cotyledon albinism [24]. The *MaOTPs*, which are chloroplast editing factors, were all up-regulated in mulberry plants, directly or indirectly affecting chloroplast development (Table 3 and Appendix A) [25]. The mutants of *FLN1*, *FLN2*, and *TRXz* resulted in leaves with yellow or albino pigmentation and caused chloroplast developmental defects through transcriptional regulation of PEP, whose functions were highly conservative in *Arabidopsis*, rice, tomato, and pepper [26,27,28]. The expression levels of *MaFLN1*, *MaFLN2*, and *MaTRXz* genes all changed significantly in mulberry trees (Table 3 and Appendix A). Therefore, according to the high conservation of functions, we speculate that these genes also play important roles in mulberry chloroplast development. 

*MaRbcL* was significantly down-regulated at 10:00 compared to 12:00 in E2 (Appendix A). The chloroplast gene *RbcL* encodes a large chain of ribulose diphosphate carboxylase. RuBisCO is made up of eight large subunits (RbcL) and eight small subunits (RbcS) [42]. The low catalytic activity of Rubisco, a key photosynthetic enzyme responsible for CO2 fixation, is one of the most important limiting factors in photosynthesis [42,43]. *Pgp064* was significantly up-regulated in E2 compared to H32 at 12:00 and 10:00 compared to 12:00 in E2 (Appendix A), encoded by the *psaA* gene, which binds to P700, the main electron donor of PSI, and electron acceptors A0, A1, and FX, and participates directly in photosynthesis [44]. The psbC (the CP43 inner antenna protein) played an important part in photosystem II (PSII), which is normally co-transcribed with psbD [45], and their expression level was significantly changed in our study. Finally, we speculate that the functions of these DEGs are similar to those of *Arabidopsis* and could affect the photosynthetic rate to further affect the yield of mulberry.

### 3.3. Mulberry MaCLA1, MaTHIC, and MaPKP2 Regulatory Network

After RNA-Seq analysis, we screened some photosynthetic- and chloroplast-related DEGs, most of which have been studied in *Arabidopsis thaliana*, but their functional mechanism in mulberry is uncertain. Here, *MaCLA1*, *MaTHIC*, and *MaPKP2* were selected for cloning and silenced in mulberry using VIGS technology to further study their functions and regulatory networks. We found that the leaves of silenced plants with *MaCLA1* and *MaTHIC* were yellow or even white, while *MaPKP2*-silenced plants exhibited some degree of leaf curling and shrinkage, showing what affects the photosynthetic efficiency of mulberry leaves (Figure 5). Compared to the control, the expression levels of silenced plants were reduced (Figure 5H). The chlorophyll a in all silent plants did not change, but the content of chlorophyll b decreased, which resulted in the total chlorophyll content being lower than that of the control plants, resulting in restricted plant growth (Figure 5I). Our results are consistent with the phenotype in *Arabidopsis* [21,22,23]. Changed genes in mulberry or *Arabidopsis* both cause abnormal growth in plants. Furthermore, due to the albino phenotype and high conservation in plant species [46,47], *CLA1* has currently been identified as a visual marker gene for VIGS in various plants, such as *Arabidopsis* [48], cotton [49], and *Hibiscus hamabo* [46]. Our study indicated that *MaCLA1* can also be used as a marker for application in gene functional studies in mulberry. In addition, we found that *MaCLA1* is co-expressed with *YSL3*, *BOU*, and *CHUP1* (Figure 6, Appendix A). *BOU* is an amino acid transporter, and *bou* mutations in *Arabidopsis* altered sulfur assimilation, carbon, and nitrogen metabolic crosstalk during plant photorespiration, and stunted plant growth [30]. CHUP1 encoded a chloroplast-localized protein, the maximum quantum yield of PSII was reduced in *Arabidopsis* mutants *chup1*, and net CO_2_ assimilation was biochemically rather than photochemically limited [31]. *MaTHIC* is co-expressed with PPR and *FLN1* (Figure 6, Appendix A). The *EMB2794* mutation in *Arabidopsis* results in a strong embryogenesis defect [32]. *MaPKP2* is co-expressed with glycosyl hydrolase 9B7 (*GH9B7*) and *GH3.1* gene encoded an IAA-amide synthase-like protein (Figure 6, Appendix A) and encodes a 3-phosphoglycerate dehydrogenase named *EDA9*, whose down-regulated *EDA9* expression causes drastic developmental defects for embryos and pollen [33]. From above, we concluded that three genes and their co-expressed genes are fundamentally involved in plant photosynthesis and chloroplast development. In addition, the three genes involved common interacting proteins in *Arabidopsis* and mulberry. CLA1 interacted with RPE, GPS1, and DXR proteins. CLA1 and DXR are the first two enzymes of the plastidial methylerythritol phosphate (MEP) pathway, which employs many compounds linked to photosynthesis and affects essential plant development [50,51]. THIC interacted with TH1, PUR5, and THI1; THIC, TH1, and THI1 are key enzymes in thiamine synthesis, which are essential for living organisms, and one mutant of which displayed thiamine auxotrophs that resulted in yellow or white leaves [52,53,54]. PKP2 interacted with ENOC, PDH-E1_ALPHA, and PGI1 (Figure 6; Appendix A). All of these genes are related to chloroplast and photosynthesis according to the annotated gene function. We concluded that three genes have relatively conserved functions in plants and can regulate plant growth by combining with multiple proteins to perform their functions. The result showed that the regulatory network of *MaCLA1*, *MaTHIC*, and *MaPKP2* plays an important role in photosynthetic processes. However, these interactive networks in mulberry are yet to be examined using molecular biology techniques.

## 4. Materials and Methods

### 4.1. Plant Materials and Growth Conditions

The three mulberry cultivars E1, E2, and H32 were selected from the Hubei Academy of Agricultural Sciences, the Hubei Academy of Agricultural Sciences (Variety), and the Zhejiang Academy of Agricultural Sciences, respectively. The plants were cultivated in the middle stem and planted in 1996 with a row spacing of 133 cm × 67 cm. The test plots with flat ground and uniform soil fertility were selected in the Hubei Province Mulberry Germplasm Resource Nursery to be repeated three times, and three mulberry cultivars with the same stem circumference, crown diameter, and tree vigor were selected as test materials in each plot. Fertilization and water management were carried out following the cultivation methods of high-yielding mulberry fields. At the same time, the prevention and control of mulberry diseases, insect pests, and bud thinning have been strengthened to ensure good leaf production. The test field soil is a typical yellow-brown loam soil with moderate fertility, slightly acidic with 5.6–6.5 pH, and medium to high organic matter content. 

### 4.2. Measurement of Net Photosynthetic Rate (Pn), Extraction of Transcriptomes, and Sequencing

From each mulberry cultivar (E1, E2, and H32), three plants with equal growth potential were selected to measure the net photosynthetic rate. Every 2.0 h from 6:00 to 18:00, leaves at positions 4, 5, 6, 7, and 8 were chosen for measurement, and 5-7 new shoots were measured repeatedly at each period. When the diurnal changes of the leaf photosynthetic curve in E1, E2, and H32 reached their peak and bottom (10:00 and 12:00) (Appendix A), the new shoots of nine mulberry plants with the same growth were collected as one biological replicate, and three biological replicates were set up for each cultivar. A total of 18 samples were cryopreserved on dry ice and sent to Beijing Biomarker Biotechnology Co., Ltd. (Beijing, China). The backup samples were placed in liquid nitrogen cryopreservation. Total RNA was extracted using Trizol Reagent, and the paired-end libraries with insert sizes of ~300 bp were constructed according to the manufacturer’s instructions and sequenced on an Illumina HiSeq 2000 platform using the PE150 mode.

### 4.3. Gene Structure and Transcriptome Mapping Re-Annotation 

Fastp [55] was used to remove adaptors, low-quality, and unknown base N which exceed the sequencing length by 10% to generate high-quality clean reads, which were mapped to the *Morus alba* reference genome [18] with Hisat2 [56]. The alignment result was compressed and sorted into the BAM file using SAMtools [57]. To perform genome re-annotation, the flowing steps were applied: (1) Stringtie [58] was used to assemble the transcriptome through the BAM file to obtain the gene structure annotation file (GTF); (2) the trinity [59] was used to execute Genome-guided assembly based on BAM file; (3) since the reference genome does not contain all gene sequences of the sequenced samples, trinity was used to perform de novo assembly for clean data; (4) the above three assembly results were integrated using PASA [60] to obtain a complete transcriptome assembly result; (5) the reviewed protein sequence of the plant from SwissPort and the *Morus notabilis* protein sequence aligned to *Morus alba* genome using GenomeThreader [61] to obtain the gene structure annotation file; and (6) EVidenceModeler [62] was used to integrate the above annotation files based on the weight value to obtain the final gene structure annotation file. To assess the quality of re-annotated genes, BUSCO [63] was used to calculate the completeness of annotated genes based on conserved genes in plants. Simultaneously, the newly annotated genes were annotated for function, GO, and KEGG using eggNOG-mapper [64].

### 4.4. Differential Expression Analysis

Based on the BAM file above, the raw count of each re-annotated gene was counted using featureCounts [65] and expression was estimated using the fragments per kb per million fragments (FPKM) method [66] to eliminate the effects of gene size and sequencing depth. Principal component analysis (PCA) was used to assess the plausibility of biological replication to ensure the accuracy of subsequent analysis results. DESeq2 [67] was used to identify DEGs between E1, E2, and H32 at 10:00 and 12:00 using a quantitative gene expression matrix (counts matrix). A DEG is defined as a gene with a |log2 (Fold Change)| ≥ 1 and a false discovery rate less than 0.05 (FDR < 0.05) [68].

### 4.5. GO and KEGG Enrichment Analysis

For Gene Ontology (GO) enrichment analysis, all DEGs were mapped to the GO database (http://geneontology.org/, accessed on 14 December 2021), and the number of genes for each GO term was calculated. Then the hypergeometric test was applied to find the significantly enriched GO terms, with a *p*-value ≤ 0.05 as the criterion. The annotation and enrichment analysis of the Kyoto Encyclopedia of Genes and Genomes (KEGG) pathway was performed using KOBAS3.0 (http://kobas.cbi.pku.edu.cn/kobas3/, accessed on 14 December 2021), with a *p*-value ≤ 0.05 as the significantly enriched criterion.

### 4.6. Virus-Induced Gene Silencing (VIGS) of Candidate Genes

Before constructing VIGS vectors, we first designed the primers using Primer 3 (https://primer3.ut.ee/, accessed on 25 December 2021) (Appendix A). A short fragment ranging from 268 bp to 355 bp (Appendix A) was cloned from the cDNA of mulberry leaf based on the respective candidate gene sequences by PCR and electrophoretic detection. Short fragments were recombined into *EcoRI*-linearized pTRV2 vectors which were linearized using the single enzyme *EcoRI* for digestion (Appendix A). Meanwhile, a *MaPDS* fragment was also recombined into *EcoRI*-linearized pTRV2 vectors, as a positive control. The homologous recombination procedure referred to the instructions of ClonExpress II One Step Cloning Kit (Vazyme, Nanjing, C112-01). All vectors including pTRV1, pTRV2, pTRV2-*MaPDS*, pTRV2-*MaCLA1*, pTRV2-*MaTHIC*, and pTRV2-*MaPKP* were introduced into *Escherichia coli*, respectively. Randomly 3–5 positive colonies were sent to the company for sequencing (Appendix A). The sequencing results are essentially consistent with the reference sequence, indicating that the vectors were successfully constructed. Then constructed vectors were transformed into *Agrobacterium tumefaciens* GV3101 for infection of mulberry cotyledons. *Agrobacterial* suspensions were adjusted to OD_600_ = 0.5, and the pTRV1 and pTRV2 suspensions were mixed in equal parts. Wild-type seeds after 15 days were used for inoculation. Agrobacterial suspensions were infiltrated into the underside of the cotyledons. After inoculation, the plants were cultivated in the dark for 3 days and then transferred to the growth chamber. Plant phenotype changes were observed and photographed at the seedling age of 40 days. 

### 4.7. Quantitative Reverse-Transcribed PCR (qRT-PCR) Analysis 

Total RNA from the infection area of VIGS plants was extracted using TRIzol reagent, and one microgram of total RNA was reverse transcribed using HiScript II 1st Strand cDNA Synthesis Kit (Vazyme) according to the supplier’s instructions. A SYBR Green-based qRT-PCR was performed on a QuantStudio 7 platform (Thermo Fisher Scientific; Shanghai, China) with a 10 μL reaction system and were used as internal controls. All primer sequences for qRT-PCR are shown in Appendix A Cycle conditions were as follows: pre-denaturation at 95 ℃ for 1 min; 95 °C 10 s; 58 °C 15 s; 72 °C 20 s, 40 cycles, followed by a standard melting curve analysis. The data were processed using the 2^−ΔΔCT^ method [69]. 

### 4.8. Measurement of Chlorophyll Content

The leaves from the infection area of VIGS plants were taken and ground with liquid nitrogen, and 0.1 g of powder was used for chlorophyll extraction. Three biological replicates were made for each genotype. Chlorophyll was extracted with 1 mL of 95% ethanol at 4 °C for 12 h under darkness and then centrifuged at 15,000 rpm for 10 min. Using 95% ethanol as a blank, the absorbance value of chlorophyll extract at 646 nm and 663 nm was measured with a microplate reader (BMG Labtech, Ortenberg, Germany), and the calculation of chlorophyll content was described in the published work [70]. 

### 4.9. Analysis of Gene Co-Expression and PPI Networks

The Weight Gene Co-expression Network Analysis (WGCNA) can calculate the similarity coefficient between all genes according to the topological overlap matrix (TOM) to construct a gene co-expression network. The R package WGCNA was performed according to the reference, adopting the dynamic hybrid tree cut algorithm to analyze RNA-Seq data with parameters of genes FPKM ≥ 1 [71]. Gene co-expression network maps were drawn using Cytoscape software [72]. The STRING online tool [73] was used to analyze the protein-protein interaction (PPI) network of genes in *Morus notabilis* and *Arabidopsis thaliana*, with a confidence threshold > 0.4. Blastp was used to convert gene ID from *Morus alba* L. to its homologous in *Arabidopsis thaliana* [74].

## 5. Conclusions

In our study, we totaled 6587 DEGs, of which 142 DEGs are involved in chloroplasts and photosynthetic genes. *MaCLA1*, *MaTHIC*, and *MaPKP2* have similar functions in *Arabidopsis* and mulberry, which are mainly involved in the regulation of mulberry photosynthesis and contribute to photosynthesis. Moreover, *MaCLA1* can be used as a marker gene for mulberry VIGS study. In the future, our study will overexpress the three genes in mulberry through a stable genetic transformation as an opportunity to increase the photosynthetic rate of mulberry leaves to increase their yield. In addition, the three genes potentially cooperate with many other genes linked to similar functions to regulate mulberry photosynthesis and chloroplast development. This study preliminarily analyzed the mulberry photosynthetic regulatory network at the transcriptomic level and provided a theoretical basis and technical support for the development of the mulberry industry.

## Figures and Tables

**Figure 1 ijms-23-08620-f001:**
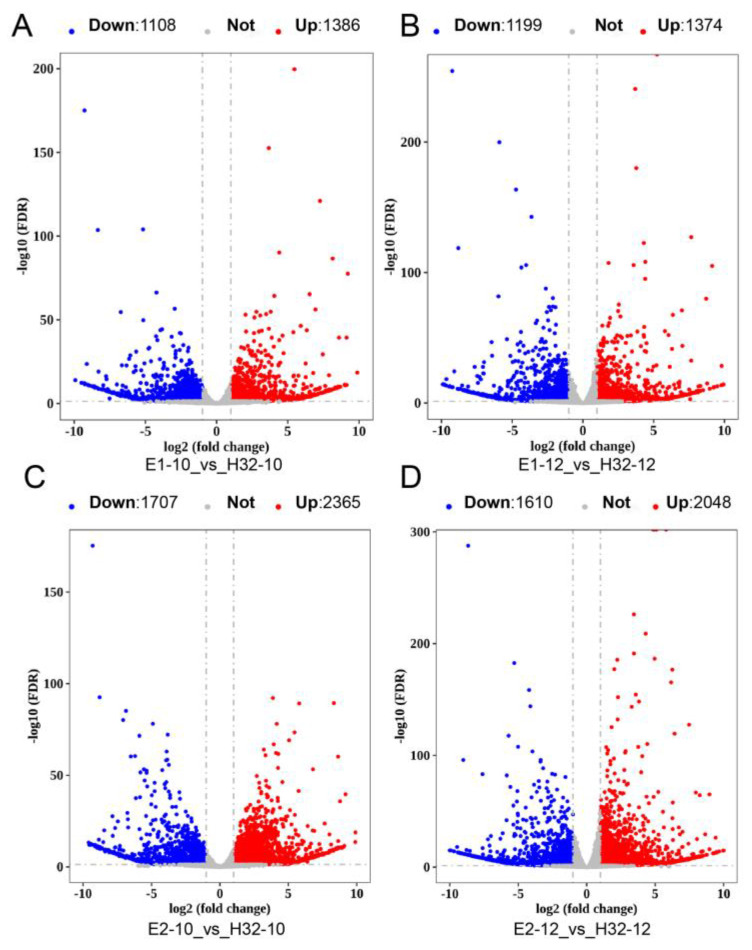
Volcano map of DEGs. (**A**) Volcano map of DEGs in E1-10_vs_H32-10; (**B**) Volcano map of DEGs in E1-12_vs_H32-12; (**C**) Volcano map of DEGs in E2-10_vs_H32-10; (**D**) Volcano map of DEGs in E2-12_vs_H32-12; The red, blue, and gray dots represent up-regulated DEGs (Up), down-regulated DEGs (Down), and not change genes (Not), respectively.

**Figure 2 ijms-23-08620-f002:**
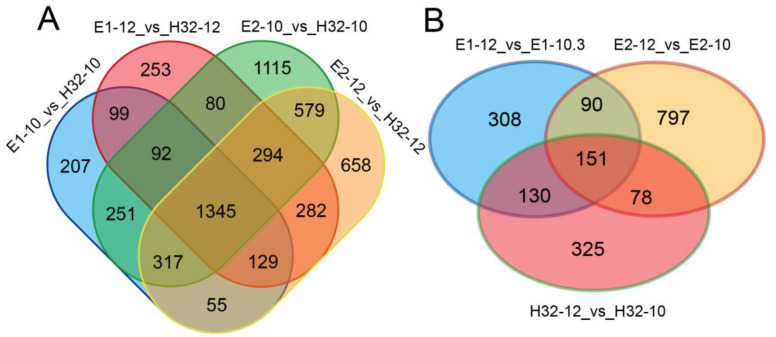
Venn diagram of DEGs. (**A**) Venn diagram of DEGs in different cultivars (**B**) Venn diagram of DEGs between 10:00 and 12:00.

**Figure 3 ijms-23-08620-f003:**
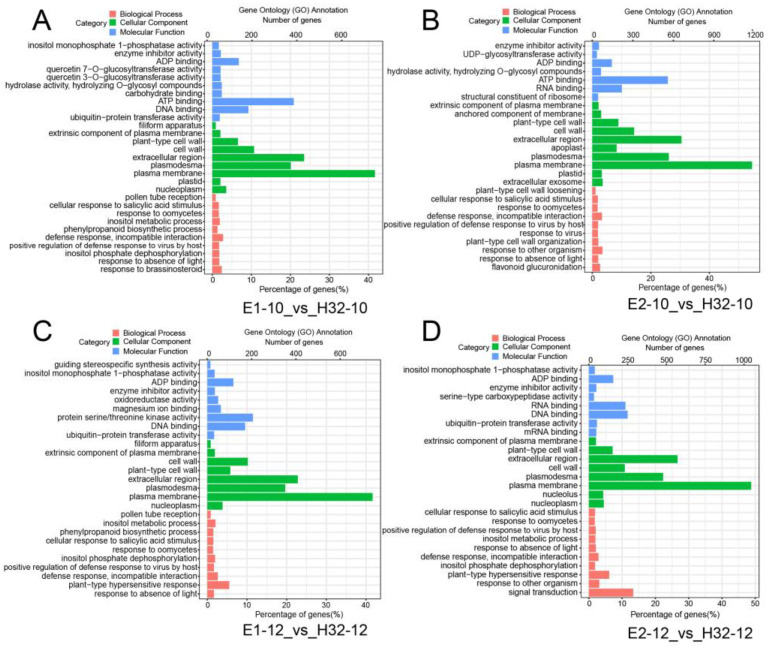
Gene ontology (GO) analysis of DEGs. (**A**) GO map of DEGs in E1-10_vs_H32-10; (**B**) GO map of DEGs in E1-12_vs_H32-12; (**C**) GO map of DEGs in E2-10_vs_H32-10; (**D**) GO map of DEGs in E2-12_vs_H32-12; y-axis represents the GO classification, the bottom and top of the x-axis represent the percentage and number of genes, respectively.

**Figure 4 ijms-23-08620-f004:**
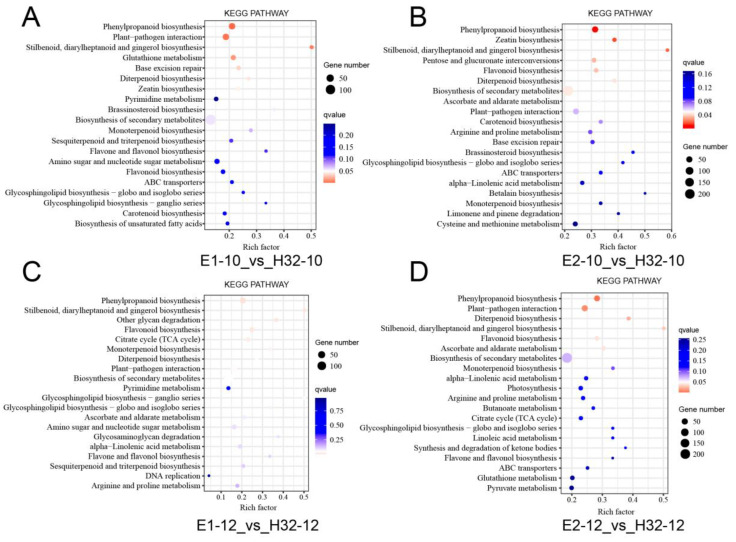
KEGG (Kyoto Encyclopedia of Genes and Genomes Pathway) analysis of DEGs. (**A**) GO map of DEGs in E1-10_vs_H32-10; (**B**) GO map of DEGs in E1-12_vs_H32-12; (**C**) GO map of DEGs in E2-10_vs_H32-10; (**D**) GO map of DEGs in E2-12_vs_H32-12. y-axis represents the pathname, and the x-axis represents the enrichment factor. The larger the enrichment factor, the higher the enrichment level of DGEs in the pathway. The *p* value is expressed by color with white as the border. The deeper the red color, the more important the accumulation of DEGs in the signaling pathway. The number of DEGs in a pathway was expressed by the size of the circle. The larger the circle, the higher the enrichment of the pathway.

**Figure 5 ijms-23-08620-f005:**
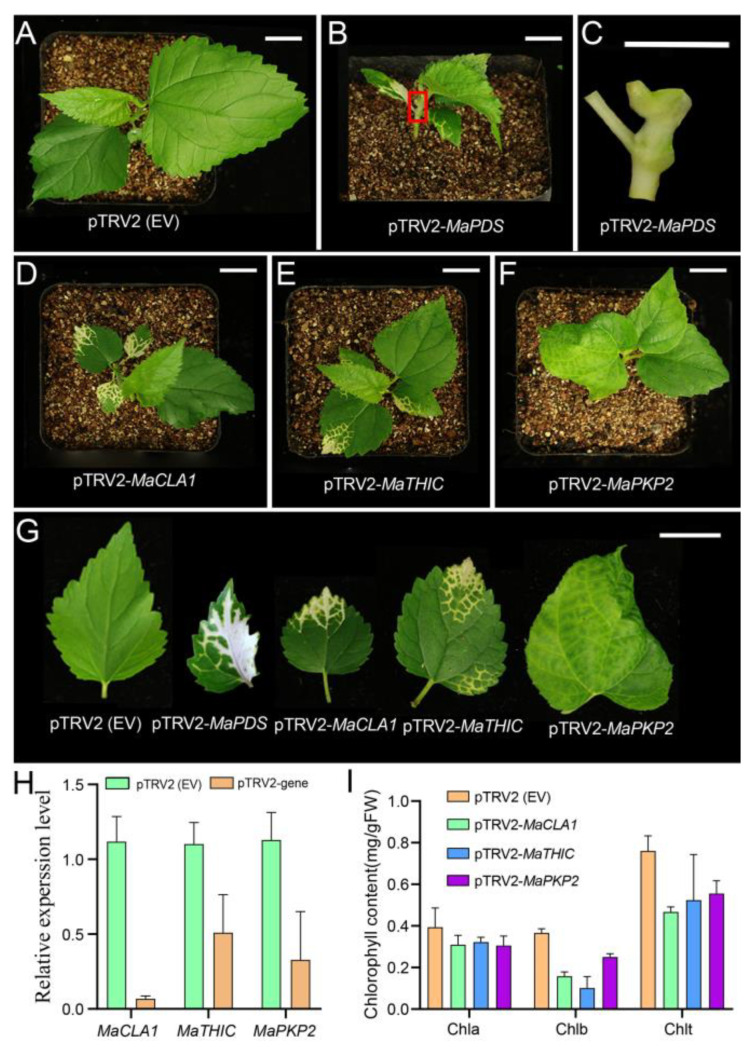
Virus-induced gene silencing (VIGS) of *MaCLA1*, *MaTHIC*, and *MaPKP2* in mulberry. (**A**) pTRV2 (EV) plant as negative control; (**B**) VIGS plant of *MaPDS* as positive control; (**C**) The stem of the enlarged map for the red box in (**B**,**D**) VIGS plant of *MaCLA1*; (**E**) *MaTHIC* VIGS plant; (**F**) VIGS plant of *MaPKP2*; (**G**) leaf phenotype of VIGS plant; Scale bar, 1 cm. (**H**) Relative expression of *MaCLA1*, *MaTHIC*, and *MaPKP2*; (**I**) Chlorophyll contents of the chlorotic leaves from VIGS plants. Error bars indicate the standard deviation (n = 2). Chla: chlorophyll a, Chlb: chlorophyll b, Chlt: total chlorophyll.

**Figure 6 ijms-23-08620-f006:**
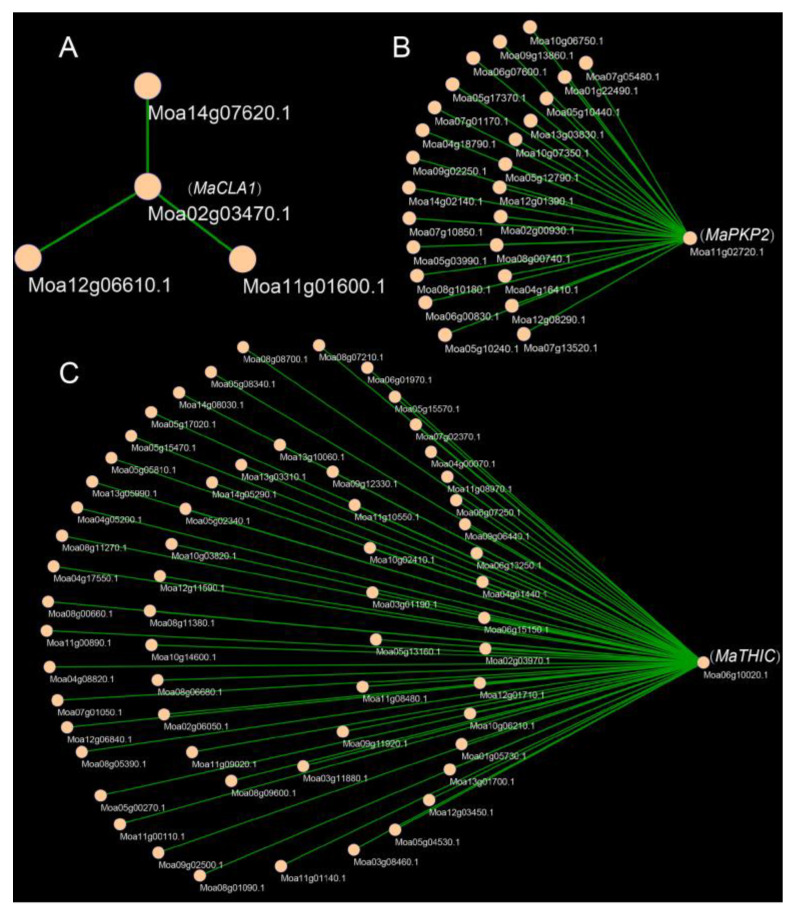
Co-expression genes of *MaCLA1*, *MaPKP2*, and *MaTHIC*. (**A**) Co-expression genes of *MaCLA1*; (**B**) Co-expression genes of *MaPKP2*, (**C**) Co-expression genes of *MaTHIC*.

**Table 1 ijms-23-08620-t001:** The statistics of alignment between transcriptome data and the reference genome of cultivated mulberry.

Sample	Total Reads	Mapped Reads	Unique-Mapped Reads	Multi-Mapped Reads	Total Alignments (%)	Unique Alignments (%)	Multi Alignments (%)
E1-10-1	49904388	45571463	44143546	1427917	91.32	88.46	2.86
E1-10-2	56420992	51638586	49910614	1727972	91.52	88.46	3.06
E1-10-3	54262738	49480593	47802340	1678253	91.19	88.09	3.09
E1-12-1	59327236	54142095	52301878	1840217	91.26	88.16	3.1
E1-12-2	50923264	46383918	44886044	1497874	91.09	88.14	2.94
E1-12-3	56418276	51324742	49280862	2043880	90.97	87.35	3.62
E2-10-1	55613350	51299112	49545826	1753286	92.24	89.09	3.15
E2-10-2	47755332	43994904	42473230	1521674	92.13	88.94	3.19
E2-10-3	57610636	53335418	51765800	1569618	92.58	89.85	2.72
E2-12-1	53042616	48929692	46449439	2480253	92.25	87.57	4.68
E2-12-2	63794488	59118053	56404894	2713159	92.67	88.42	4.25
E2-12-3	51907882	47912048	45849994	2062054	92.3	88.33	3.97
H32-10-1	45519424	42730393	41251247	1479146	93.87	90.62	3.25
H32-10-2	53754266	50627556	48978022	1649534	94.18	91.11	3.07
H32-10-3	51907008	48613770	47046841	1566929	93.66	90.64	3.02
H32-12-1	53312124	50020860	48357132	1663728	93.83	90.71	3.12
H32-12-2	53322894	50059674	48425378	1634296	93.88	90.82	3.06
H32-12-3	57476448	53999036	52269167	1729869	93.95	90.94	3.01

**Table 2 ijms-23-08620-t002:** Assessment for gene structure re-annotation of *Morus alba*.

Type	*Morus alba*
Number of genes	22,664
Mean length of genomic loci	3213
Mean exon number	5.07
Mean CDS length	1226
Genes with functional annotations	20,185
Genes with GO terms	12,438
Gene with ko terms	10,379
Complete BUSCOs	93.40%
Fragmented BUSCOs	4.20%
Missing BUSCOs	2.40%

**Table 3 ijms-23-08620-t003:** Important genes related to chloroplasts and photosynthesis.

Gene_ID	At_Gene	Identify (%)	Gene_Name	Function
Moa12g01070.1	AT2G37770.2	68.932	*ChlAKR*	NAD(P)-linked oxidoreductase superfamily protein
Moa02g03470.1	AT4G15560.1	87.593	*CLA1*	Deoxyxylulose-5-phosphate synthase
Moa02g13070.1	AT1G15510.1	65.862	*ECB2*	Tetratricopeptide repeat (TPR)-like superfamily protein
Moa07g15800.1	AT3G18110.1	75.309	*EMB1270*	Pentatricopeptide repeat (PPR) superfamily protein
Moa11g02010.1	AT3G49240.1	62.937	*emb1796*	Pentatricopeptide repeat (PPR) superfamily protein
Moa12g01710.1	AT3G54090.1	70.41	*FLN1*	fructokinase-like 1
Moa14g05970.1	AT1G69200.1	63.729	*FLN2*	fructokinase-like 2
Moa02g05740.1	AT3G23020.1	60.093	*PPR30*	Tetratricopeptide repeat (TPR)-like superfamily protein
Moa09g08310.1	AT1G01320.2	60.808	*REC1*	Tetratricopeptide repeat (TPR)-like superfamily protein
Moa10g03540.1	AT5G39980.1	83.251	*EMB3140*	Tetratricopeptide repeat (TPR)-like superfamily protein
Moa07g06690.1	AT5G59200.1	64.516	*OTP80*	Tetratricopeptide repeat (TPR)-like superfamily protein
Moa06g09830.1	AT2G29760.1	60.377	*OTP81*	Tetratricopeptide repeat (TPR)-like superfamily protein
Moa01g09900.1	AT1G08070.1	66.584	*OTP82*	Tetratricopeptide repeat (TPR)-like superfamily protein
Moa04g00100.1	AT3G57430.1	65.682	*OTP84*	Tetratricopeptide repeat (TPR)-like superfamily protein
Moa14g04910.1	AT2G02980.1	67.295	*OTP85*	Pentatricopeptide repeat (PPR) superfamily protein
pgp033	ATCG00730.1	98.758	*petD*	photosynthetic electron transfer D
Moa11g02720.1	AT5G52920.1	84.974	*PKP-BETA1*	plastidic pyruvate kinase beta subunit 1
pgp064	ATCG00350.1	97.867	*psaA*	Photosystem I, PsaA/PsaB protein
pgp065	ATCG00340.1	97.684	*psaB*	Photosystem I, PsaA/PsaB protein
pgp038	ATCG00680.1	97.441	*psbB*	photosystem II reaction center protein B
pgp068	ATCG00280.1	97.674	*psbC*	photosystem II reaction center protein C
pgp080	ATCG00080.1	100	*psbI*	photosystem II reaction center protein I
pgp067	ATCG00300.1	91.935	*psbZ*	YCF9
pgp056	ATCG00490.1	94.737	*rbcL*	ribulose-bisphosphate carboxylases
pgp074	ATCG00170.1	80.87	*rpoC2*	DNA-directed RNA polymerase family protein
pgp031	ATCG00750.1	92.754	*rps11*	ribosomal protein S11
Moa07g03450.1	AT3G51895.1	76.888	*SULTR3:1*	sulfate transporter 3;1
Moa06g10020.1	AT2G29630.3	86.574	*THIC*	thiaminC
Moa02g08320.1	AT3G06730.1	78.621	*TRXz*	Thioredoxin z

## Data Availability

The RNA-Seq raw reads have been deposited into the National Center for Biotechnology Information Sequence Read Archive under accession codes PRJNA836246. The re-annotation results can be downloaded from figshare: https://figshare.com/articles/dataset/Re-annotation_of_the_mulberry_genome/19729525, accessed on 8 May 2022.

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
