# Peer review of "Screening and Verification of Photosynthesis and Chloroplast-Related Genes in Mulberry by Comparative RNA-Seq and Virus-Induced Gene Silencing"

_ijms, 2022, doi:10.3390/ijms23158620_

Round 1
Reviewer 1 Report
l28-29 the link between results summarized in the abstract and the last sentence is not clear and hardly makes sense in this context
l44 english seems to be not correct, please check
l70-71 this statement is trivial and superfluous
l74-77 I don't see a connection to the focus of the study
l93 a more detailed explanation of the choice of harvest times is required, does this mean 10:00 or 12:00? Why is the timepoint important for the investigation?
l94-101 this should be better part of the abstract
l101-104 as the authors didn't perform any measurements on photosynthesis (except chlorophyll content) or investigations on biomassproduction, this statement is out of the focus of the present study.
Tables: e.g. Table S1: better choose landscape format, because the tables (headlines in grey) aren't well to read
Explanation of the tables is missing e.g. table S4, what do the red lines mean?
l293-300 method
Discussion 3.1: Sorry, but this is known already for a long time...so what is new in the context of your study?
Reviewer 2 Report
Review of the manuscript entitled: Screening and verification of photosynthesis and chloroplast related genes in mulberry by comparative RNA-Seq and virus-induced gene silencing.
The authors identified the differentially expressed genes in two high-yielding and one low–yielding cultivars of mulberry. The experiment was performed at two time points described as 10:00 and 12:00. Additionally, the authors performed research on mulberry plants with silence genes: MaCLA1, MaTHIC, MaPKP2.
The performed experiments resulted in obtaining vast amounts of data. The experiments were planned and carried out correctly, however, a few issues should be clarified by the authors. What do time points 10:00 and 12:00 mean? Why were they chosen? Also why the authors decide to silence these particular genes remain unclear. The way the results are presented is illegible. What are more the results mostly contain statistical information on the analyzes performed (Why were additional materials included in the text?).
The obtained results have not been sufficiently analyzed in the context of the knowledge already available and in the present form does not bring any scientific novelty. On the other hand, some conclusions are too speculative and not supported by results. For example: “Highly active Rubisco may also have the potential to improve photosynthetic efficiency and plant productivity” (p.12 line 266-267) – RubisCO is a large protein complex the activity of which is regulated by many factors and divergent processes. The increase in expression of the gene encoding the large subunit of the RubisCO is an insufficient premise to conclude about the increased activity of RubisCO.
The descriptions of the proteins encoded by some of the described genes are inaccurate and must be corrected. For example” PSII CP43 reaction center protein, encoded by psbC gene (Table 3, Table S4), which is a novel chlorophyll protein complex that affected photosynthesis by participating in chlorophyll synthesis”(p12 line 274-276). The PsbC protein is well-known core protein of PSII center, an apoprotein of a complex, forming inner antenna of PSII. Recently discovered involvement in chlorophyll synthesis is just its additional role of PsbC.
Or
The above results indicated that the core proteins psaA, psbC, and rbcL in PSI and PSII were highly conserved in plants ( p.12 lines 277- 279) – the rbcL is not a photosystem protein.
Why were these fragments placed in a “result” section not in “discussion”?
The results obtain for MaCLA1 and MaTHIC are confirmation of already known information.
The discussion is unclear and of too general a level. Some information are not related to results, for example: “Highly conserved in plants, RbcL translocates plastids to the nucleus to achieve functional complementation, thereby improving yield and using of nitrogen efficiency to increase biomass production” (p.34, lines432-434)
Other conclusions refer to basic knowledge: “The above results indicated that the photosynthetic- and chlorophyll-related gene mutation led to changed gene expression to affect the photosynthetic rate and yield of the plant, and hinder the growth and development of the plant” ( p.34.line440-442)
In conclusion. The results obtained by the authors certainly have publishing potential. However, they need to be analyzed more carefully, described more clearly, and set correctly in a physiological context. The manuscript in its present form should be rejected.
Round 2
Reviewer 2 Report
The authors responded to most of my comments and remarks, but the manuscript still needs to be revised. Unfortunately, in the new version of the manuscript, some corrections are illegible (for example lines 96-104, 487-497, 592, or 641). The authors did not incorporate the information about the reasons for choosing these two particular time points in the text. This is important information and should be supplemented.
The language of the manuscript needs to be corrected. There are also other issues related to describe genes that are inaccurate and need to be improved. The authors should pay much more attention to the correctness of the wording used. The meaning of some terms in the scientific language is very precise.
The sentence “In addition, the expression levels of FLN1, FLN2, and TRXz, which are involved in PEP transcription regulation changed significantly in this study” (p.11 lines 293-294) needs to be corrected. PEP stands for plastid-encoded RNA polymerase and above mention factors, FLN1, FLN2, and TRXz regulate the transcriptional activity of PEP, not the transcription of PEP.
The sentence “However, all plant growth and development are the results of the regulation of gene expression, while photosynthesis is the basic life activity for organisms.”( p.31 lines 429-430) needs to be rephrased since it is unclear and questionable. The regulation of plant growth and development is a very complex matter that involved many processes both upstream and downstream from the regulation of gene expression.
The sentence “The psbC was a novel chlorophyll protein complex in PSII, which affected photosynthesis by participating in chlorophyll synthesis” (p.32 line 502-503) must be rephrased. The cited work is not referred to chlorophyll synthesis but photosystem II assembly.
The sentence “The result showed gene participate in the regulation of photosynthesis through the interaction of various photosynthetic processes” ( p.33. lines 549-550) must be rephrased. In the regulation of photosynthesis participate proteins encoded by given genes, not the genes themselves.
Why are conlusions placed after the “ Methods” section?
“This study preliminarily analyzed the mulberry photosynthetic regulatory network at the omics and molecular level” – please rephrase the sentence to “This study preliminarily analyzed the mulberry photosynthetic regulatory network at the transcriptomic level”
Author Response
Dear Editor-in-Chief,
Thank you for the valuable comments on our paper again. Based on the comments, we have
made corresponding modifications on the first revised manuscript. Here, we would like to submit
our newly revised version for your approval. Corresponding changes in the newly revised
manuscript are highlighted in blue. The point-by-point responses to the comments are listed
below.
We look forward to your positive response!
Sincerely
Yong Li
1. The authors responded to most of my comments and remarks, but the manuscript still
needs to be revised. Unfortunately, in the new version of the manuscript, some corrections
are illegible (for example lines 96-104, 487-497, 592, or 641). The authors did not incorporate
the information about the reasons for choosing these two particular time points in the text.
This is important information and should be supplemented.
Response: We have supplemented the net photosynthetic rate (Pn) data for two high-yielding
(E1, E2) and one low-yielding (H32) mulberry genotype in Figure S1, and the corresponding
context has been added in “Results” and “Methods” with blue font in line 106-114 and line 471-
475, respectively.
2. The language of the manuscript needs to be corrected. There are also other issues related
to describe genes that are inaccurate and need to be improved. The authors should pay much
more attention to the correctness of the wording used. The meaning of some terms in the
scientific language is very precise.
Response: We have checked and revised the entire article again, all changes (including syntax and
wording used) are marked in blue.
3. The sentence “In addition, the expression levels of FLN1, FLN2, and TRXz, which are
involved in PEP transcription regulation changed significantly in this study” (p.11 lines 293-
294) needs to be corrected. PEP stands for plastid-encoded RNA polymerase and above
mention factors, FLN1, FLN2, and TRXz regulate the transcriptional activity of PEP, not the
transcription of PEP.
Response: Thank you for pointing out this issue, we have changed it to “Furthermore, the
expression levels of FLN1, FLN2 and TRXz, which regulated the transcriptional activity of plastid-
encoded RNA polymerase (PEP) were significantly altered in this study” in line 246-248
4. The sentence “However, all plant growth and development are the results of the regulation
of gene expression, while photosynthesis is the basic life activity for organisms.” ( p.31 lines
429-430) needs to be rephrased since it is unclear and questionable. The regulation of plant
growth and development is a very complex matter that involved many processes both
upstream and downstream from the regulation of gene expression.
Response: Thank you for reporting the issue, the sentence has been changed to “However, the
result of gene expression regulation is one of the factors affecting photosynthesis to further
impact plant growth and development” in lines 354-355.”
5. The sentence “The psbC was a novel chlorophyll protein complex in PSII, which affected
photosynthesis by participating in chlorophyll synthesis” (p.32 line 502-503) must be
rephrased. The cited work is not referred to chlorophyll synthesis but photosystem II
assembly.
Response: Thank you for pointing out the issue, we have changed it to “The psbC (the CP43 inner
antenna protein) played an important part in photosystem II (PSII), which normally co-
transcribed with psbD, and their expression level as significantly changed in our study.” in line
404-406
6. The sentence “The result showed gene participate in the regulation of photosynthesis
through the interaction of various photosynthetic processes” ( p.33. lines 549-550) must be
rephrased. In the regulation of photosynthesis participate proteins encoded by given genes,
not the genes themselves.
Response: Thank you for pointing out this issue, we reformulated “The result showed that the
regulatory network of MaCLA1, MaTHIC, and MaPKP2 play an important role in photosynthetic
processes.” on lines 441-452 in the newly revised manuscript
7. Why are conlusions placed after the “Methods” section?
Response: Thank you, we set the layout of the text according to the template given by Int. J. Mol.
Sci, which conclusion is placed after the “methods” section.
8. “This study preliminarily analyzed the mulberry photosynthetic regulatory network at the
omics and molecular level” – please rephrase the sentence to “This study preliminarily
analyzed the mulberry photosynthetic regulatory network at the transcriptomic level”
Response: Thank you for the suggestion, we have rephrased it.
